# Coffee Consumption and Its Inverse Relationship with Gastric Cancer: An Ecological Study

**DOI:** 10.3390/nu12103028

**Published:** 2020-10-02

**Authors:** Luis G. Parra-Lara, Diana M. Mendoza-Urbano, Juan C. Bravo, Constain H. Salamanca, Ángela R. Zambrano

**Affiliations:** 1Facultad de Ciencias de la Salud, Universidad Icesi, Cali 760032, Valle del Cauca, Colombia; luisgabrielparralara@hotmail.com; 2Centro de Investigaciones Clínicas (CIC), Fundación Valle del Lili, Cali 760032, Valle del Cauca, Colombia; diana.mendoza@fvl.org.co; 3Department of Pathology, Fundación Valle del Lili, Cali 760032, Valle del Cauca, Colombia; juan.bravo@fvl.org.co; 4Laboratorio de Diseño y Formulación de Producto, Departamento de Ciencias Farmacéuticas, Facultad de Ciencias Naturales, Universidad Icesi, Cali 760032, Valle del Cauca, Colombia; chsm70@gmail.com; 5Department of Hemato-Oncology, Fundación Valle del Lili, Cali 760032, Valle del Cauca, Colombia

**Keywords:** coffee, antioxidants, stomach neoplasms, mortality, observational study

## Abstract

Coffee is the second most popular drink worldwide, and it has various components with antioxidant and antitumor properties. Due to its chemical composition, it could act as an antitumor substance in the gastrointestinal tract. The objective of this study was to explore the relationship between coffee consumption and the incidence/mortality of stomach cancer in the highest-consuming countries. An ecological study using Spearman’s correlation coefficient was performed. The WorldAtlas’s dataset of coffee consumption and the incidence/mortality rates database of the International Agency for Research were used as sources of information. A total of 25 countries were entered to the study. There was an inverse linear correlation between coffee consumption in kg per person per year and estimated age-adjusted incidence (*r* = −0.5984, *p* = 0.0016) and mortality (*r* = −0.5877, *p* = 0.0020) of stomach cancer. Coffee may potentially have beneficial effects on the incidence and mortality of stomach cancer, as supported by the data from each country analyzed.

## 1. Introduction

With a mean annual consumption of 500 billion cups per year, coffee is the second most popular drink in the world [1]. It is obtained through Coffea seeds that are processed with water, producing a unique mix of bioactive compounds [2]. Among its most outstanding components are substances known for their antioxidant and antitumor properties [3,4].

Antioxidants are substances with the ability to prevent, delay, or eliminate the oxidative damage of a target molecule; in the case of cells, they include proteins and nucleic acids [5,6]. According to their mechanisms of action, antioxidants are classified into three groups: (1) those that prevent the formation of new free radicals, (2) those that capture existing free radicals, and (3) those that repair the damage [7]. Due to their mechanisms of action and the role of oxidative stress in the pathogenesis of different cancers, some substances are proposed as chemo preventive agents.

Gastric cancer is one of the five most frequent malignancies, with an annual mortality rate of approximately 723,000 deaths per year [8]. Its clinical presentation has a multifactorial display involving elements such as age and eating habits [9]. The relationship between coffee consumption and gastric cancer has been a topic of research, but the findings are contradictory [2,10]. On the one hand, a meta-analysis reported that coffee consumption was associated with the development of gastric cancer [11]. In contrast, two meta-analyses did not find an association with the risk of gastric cancer [12,13]. However, confounding factors such as age, race, alcohol intake, tea consumption, and smoking were not adjusted for, nor was how differences in coffee preparation methods may affect the concentrations of different compounds.

Although there is much controversy in this regard, it is known that coffee contains chemical compounds with significant antitumoral activities that could affect the gastrointestinal tract. In this way, the question arises of whether there is a relationship between coffee consumption and the incidence/mortality of stomach cancer in the highest-consuming countries.

## 2. Materials and Methods

An ecological study was performed. The analysis used two independent sets of information, coffee consumption and cancer incidence/mortality data. The coffee consumption data in kg per person per year were available in the WorldAtlas dataset from 25 countries and published in January 2018. The estimated age-adjusted (world standard) incidence and mortality rates of stomach cancer per 100,000 inhabitants were obtained from the International Agency for Research on Cancer GLOBOCAN database published in 2018 (the last year when complete data were available). The research team reviewed each country’s cancer fact sheets for 2018.

Scatter plots for the age-standardized incidence and mortality rate of stomach cancer based on coffee consumption were used for statistical analysis. Spearman’s rank correlation coefficient was also calculated according to the data distribution. All data analysis was performed using STATA^®^ (Version 14.0, StataCorp L.P., and College Station, TX).

## 3. Results

### Coffee Consumption and Stomach Cancer in Different Countries

A total of 25 countries worldwide were included in the study. Table 1 summarizes their coffee drinking and incidence/mortality by stomach cancer. Finland reported higher coffee consumption and had relatively low incidences and mortality rates for gastric cancer. Portugal, Estonia, and Bosnia and Herzegovina showed an outlier distribution. These countries consumed an average of 5 kg of coffee per person per year but had an incidence of over ten and a mortality of over seven (age-standardized rates) per 100,000 inhabitants. The USA reported the lowest coffee consumption, with 4.2 kg per person per year, and had among the lowest incidences and mortality rates of gastric cancer.

There was a moderate inverse linear correlation (*r* = −0.5984, *p* = 0.0016) between coffee consumption in kg per person per year and estimated, age-adjusted incidence of stomach cancer (Figure 1). Sweden was the top performer in terms of both the incidence of stomach cancer and coffee consumption. The slope of the regression line allowed us to estimate that it would take about 7 kg of coffee per person per year to decrease the incidence of stomach cancer in a given country by a factor of six. For Norway that would amount to 9.7 kg per year; the minimally effective coffee dose seems to hover around 5–6 kg per year.

There was also a moderate inverse linear correlation (*r* = −0.5877, *p* = 0.0020) between coffee consumption in kg per person per year and estimated age-adjusted mortality due to stomach cancer (Figure 2). Sweden was also the top performer in terms of both mortality due to stomach cancer. The slope of the regression line allowed us to estimate that it would take about 7 kg of coffee per person per year to decrease the mortality of stomach cancer in a given country by a factor of four.

## 4. Discussion

Coffee is one of the most popular beverages worldwide and contains hundreds of biologically active phytochemicals, including many with antioxidant activity. Therefore, the relationships between coffee drinking and the incidence/mortality of cancer have been studied, especially in the gastrointestinal tract [10]. Thus, the potential role of coffee drinking in gastric cancer remains under study. In this study, the principal finding was an inverse linear correlation between coffee consumption per person per year and age-adjusted incidence/mortality of stomach cancer in the 25 countries with the highest coffee consumption. The described correlation does not prove causality but suggests a common underlying mechanism between the events of interest and permits us to speculate about the identity of the possible mechanism.

The inverse correlation between coffee ingestion and deaths from all causes was already described in a large, prospective cohort study in the USA [14], and in the UK, Tran and collaborators found evidence that suggested an inverse association between coffee consumption and hepatocellular carcinoma [15]. However, a review of the available evidence collected on coffee drinking and stomach cancer showed heterogeneous and inconclusive results [10,11,13]. Shen et al. [11] performed a meta-analysis to evaluate the effect of coffee consumption on the risk of gastric cancer, and found an association (*Relative Risk* = 1.24, 95% CI: 1.03–1.49). They took the pooled effect size from eight prospective cohort studies that involved 312,993 participants; however, most of the participants were men, older, smokers, and drinkers, which may have resulted in overestimation of the effect. Moreover, the coffee intake information levels were reported and collected by the volunteers themselves, which could result in biases. Finally, the review only adjusted the outcome analysis by gender, age, and smoker status and did not thoroughly investigate other interest subgroups.

Another meta-analysis included 12 prospective cohort studies with 840,651 subjects and found that coffee intake was not significantly associated with overall gastric cancer risk (*Pooled Relative Risk* = 1.12, 95% CI: 0.93–1.36) [12]. Similar findings were reported in another meta-analysis that included nine prospective cohort studies involving 1,250,825 participants (*Pooled Hazard Ratio* = 1.05, 95% CI: 0.88–1.25) [13]. Nevertheless, data about coffee consumption throughout life, type of coffee consumed, anatomical locations of gastric cancer, *Helicobacter pylori* infection, and precancerous lesions, were not considered in those studies. Prospective research is required to include the variables and clarify a possible causal association between coffee and gastric cancer. In the same way, coffee consumption is related to some factors such as alcohol consumption [16], smoking [17], obesity, and lifestyles [18], all of which are associated with increased gastric cancer risk.

Li et al. [10] developed another meta-analysis that included 20 prospective cohort studies with 1,372,811 participants. Through their subgroup analysis, an increased risk of gastric cancer for participants from the United States was observed. This subgroup comprised 500,825 participants from three studies. Another interesting finding was that European and Japanese populations did not follow the same behavior even when their coffee consumption per capita was twice as great among European countries as in the United States. Conversely, the authors suggested that coffee beverages might have a protective effect against the incidence of gastric cancer and recommended exploring the dose–response association between coffee consumption and the risk of gastric cancer in future studies.

The biological plausibility of coffee’s chemopreventive properties is supported by its biochemical composition. It contains phenolic compounds such as hydroxyhydroquinone, chlorogenic acid, and caffeic acid that contribute to the activation of antioxidant enzymes to modulate oxidative stress in direct (through radical scavenging activity) and indirect ways (through activation of the Nrf2/ARE cellular system). It can also suppress cancer growth through anti-estrogenic pathways, mitochondrial toxicity, and anti-inflammatory environmental regulation [4,19]. Coffee also contains cafestol and kahweol, two lipids with an anti-genotoxic activity that act as reactive oxygen species scavengers and prevent the effects of carcinogens such as hydrogen peroxide and (4,5-b)pyridine. Additionally, they increase the expression of enzymes involved in the detoxification of DNA-reactive metabolites (glucuronosyltransferase and glutathione S-transferase) and the DNA-repair complex enzyme system [3].

Cafestol and kahweol are mainly present as fatty esters in unfiltered coffee, and are both found in low concentrations in the filtered preparation. Filtered coffee is the most widely consumed form in Central Europe and the United States [20]. Despite that, a clinical trial found that paper-filtered coffee protects humans against oxidative DNA damage [21]. DNA damage was measured in peripheral lymphocytes in single cell gel electrophoresis assays, and a 12.3% (*p* = 0.006) decrease in DNA migration was attributable to the formation of oxidized purines after coffee intake. Researchers concluded that coffee consumption prevents the endogenous formation of oxidative DNA damage, which may be causally related to beneficial health effects.

Our study showed a lower incidence/mortality for gastric cancer in Central European countries and the USA. However, they had lower coffee consumption per capita, and as previously mentioned, the most common preparation was the filtered coffee beverage. A possible explanation of this phenomenon could be found by searching the individual-level information of each country, rather than the population level, and considering the particular illness behavior in each nation.

Another possible explanation for our results is the reverse causality hypothesis. Participants that experience early gastric cancer symptoms, such as poor appetite or heartburn, may diminish coffee consumption, increasing mortality among this group. Although it is not possible to determine causality from these findings, future randomized clinical trials could be oriented to clarify this relationship.

Our results are noteworthy in the outlier distribution of countries such as Portugal, Estonia, and Bosnia and Herzegovina, which all presented high incidence/mortality of gastric cancer beyond the expectations for the linear relationship found in this study. The high incidence could be explained in countries such as Portugal by the high prevalence of *Helicobacter pylori* infection, which caused it to be the country with the highest gastric cancer mortality rate in Western Europe [22]. Similar results have been reported in Estonia, where the five-year survival rate for gastric cancer is significantly lower than in other European countries; the explanation relies on the increased number of men, elderly, smokers, and obese people in this nation [23]. Finally, according to the Digestive Cancers Europe organization [24] in 2018, Bosnia and Herzegovina had a gastric cancer incidence of 20.5 per 100,000 inhabitants, which is one of the highest reported in Europe.

Another outlier distribution was found in the USA. It presented the lowest incidence/mortality rates of gastric cancer and the lowest coffee consumption among the analyzed countries, with numbers below the linear expectations. This is in line with the information reported at GLOBOCAN 2018 [8], where the USA had one of the lowest incidences of gastric cancer worldwide, ranking 15 out of 20 evaluated countries with an age-standardized incidence rate per 100,000 inhabitants of 5.6 for men and 2.8 for women.

There are some limitations to this ecological study. Firstly, it was conducted with secondary population-level data (countries) as the primary source of information, instead of individual data. Potential information biases could have been introduced during the development of our analysis and could limit the conclusions concerning causality. Secondly, the study did not include the distinctions among coffee preparation methods (espresso, boiled, or filtered), coffee seed types (Arabica or Canephora), and temperatures/times of roasting beans in the analyses. A considerable variability of antioxidants/antitumor concentration is dependent on the mixture preparation [25,26]. However, one study evaluated the ability of free radical scavengers and protection against DNA damage by comparing 13 different varieties of coffee roasting. It found that all the preparations exhibited similar antioxidant activity; this situation was explained by the presence of polyphenolic extract in all evaluated beverages [27,28]. Conversely, some publications have shown that the temperature and time of roasting can increase the antioxidant capacity of the coffee drink through the increase of antioxidants such as melanoidin [29,30].

Certain confounding factors such as diet, lifestyle (consumption of fruits, vegetables, red meat, salt, cigarettes, and alcohol, and physical activity), socioeconomic status, place of residence, race, health insurance, and *Helicobacter pylori* infection have been shown to play paramount roles in gastric cancer development [31,32,33,34,35]. Conversely, the data available for our ecological analysis were based on population-level information and were insufficient to propose a multilevel analysis. Finally, coffee composition is very complex, and its relationship with cancer even more so. That is why we are motivated to promote research projects in the field, which should allow the identification of new therapeutic and prevention options in gastric cancer and thus mitigate its incidence and mortality.

## 5. Conclusions

Coffee consumption could increase the bioavailability of antitumor/antioxidant substances in the gastrointestinal tract. It may potentially have beneficial effects on incidence and mortality of stomach cancer and correlates with the rates of the disease in each analyzed country.

## Figures and Tables

**Figure 1 nutrients-12-03028-f001:**
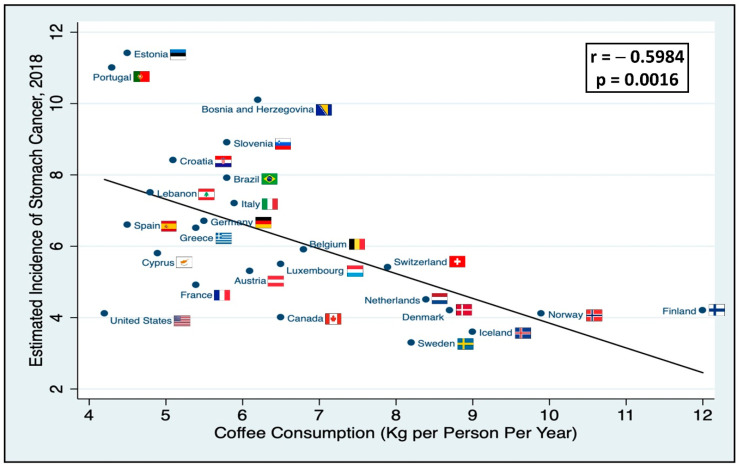
Coffee consumption and stomach cancer incidence. It shows the per country correlation between annual coffee consumption and the estimated age-standardized incidence of stomach cancer in 2018. Lineal model for incidence; incidence of stomach cancer = 10.79–0.69 * (coffee consumption).

**Figure 2 nutrients-12-03028-f002:**
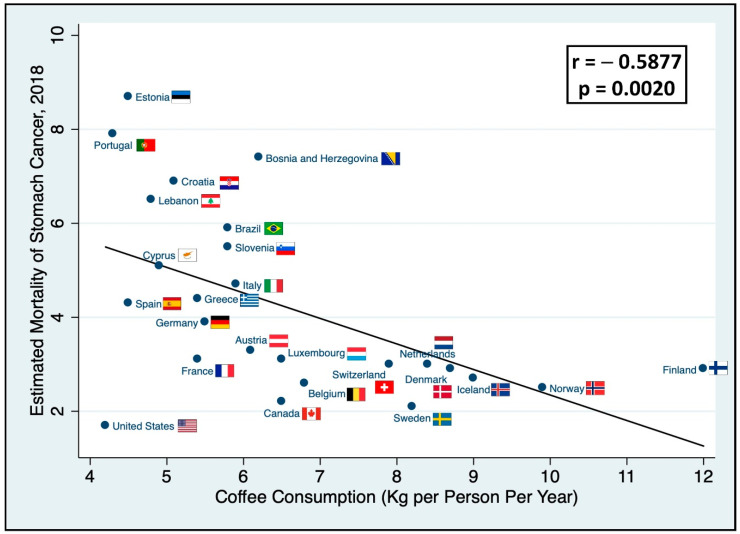
Coffee consumption and stomach cancer mortality. It shows the per country correlation between annual coffee consumption and the estimated age-standardized mortality of stomach cancer in 2018. Lineal model for mortality; mortality of stomach cancer = 7.78–0.54 * (coffee consumption).

**Table 1 nutrients-12-03028-t001:** Coffee consumption and the incidence/mortality by stomach cancer by country. It shows in descending order the coffee consumption by country and the estimated incidence/mortality of stomach cancer in 2018.

Country	Population Projection ∗	Coffee Consumption ^1^	ASR ^2^ Incidence	ASR ^2^ Mortality
Finland	5,529,000	12.0	4.2	2.9
Norway	5,381,000	9.9	4.1	2.5
Iceland	36,700	9.0	3.6	2.7
Denmark	5,832,000	8.7	4.2	2.9
Netherlands	17,431,000	8.4	4.5	3.0
Sweden	10,364,000	8.2	3.3	2.1
Switzerland	8,647,000	7.9	5.4	3.0
Belgium	11,533,000	6.8	5.9	2.6
Canada	38,116,000	6.5	4.0	2.2
Luxembourg	632,000	6.5	5.5	3.1
Bosnia and Herzegovina	3,281,000	6.2	10.1	7.4
Austria	8,926,000	6.1	5.3	3.3
Italy	60,294,000	5.9	7.2	4.7
Brazil	211,420,000	5.8	7.9	5.9
Slovenia	2,098,000	5.8	8.9	5.5
Germany	83,243,000	5.5	6.7	3.9
France	64,926,000	5.4	4.9	3.1
Greece	11,665,000	5.4	6.5	4.4
Croatia	4,105,000	5.1	8.4	6.9
Cyprus	894,000	4.9	5.8	5.1
Lebanon	6,825,000	4.8	7.5	6.5
Estonia	1,331,000	4.5	11.4	8.7
Spain	47,441,000	4.5	6.6	4.3
Portugal	10,264,000	4.3	11.0	7.9
United States	381,800,000	4.2	4.1	1.7

∗ Exponential projection of the population in July 2020; ^1^ measurement at kg per person per year; ^2^ ASR: Age-standardized rates per 100,000 inhabitants.

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
