# Peer review of "Coffee Consumption and Its Inverse Relationship with Gastric Cancer: An Ecological Study"

_nutrients, 2020, doi:10.3390/nu12103028_

Round 1

Reviewer 1 Report

Dear Authors,

I read your ecological study with great interest. I did a very short literature review and could not find any similar studies. However, I think the following suggestions could help you improve the manuscript and make it more interesting for the readers:

Major suggestions:

  • The statistical method applied doesn't allow for adjusting on known confounders of gastric cancer like age and alcohol consumption. I suggest using regression models to overcome this limitation as suggested by (https://doi.org/10.1038/sj.ebd.6400454).
  • While the results are in line with some studies (https://doi.org/10.6133/apjcn.092015.07), other studies showed non-conclusive with regard to coffee consumption’s risk on gastric cancer like (https://doi.org/10.1038/s41416-019-0465-y). This is not fully addressed in the discussion section of the manuscript.

Minor suggestions:

  • The manuscript includes some grammatical and wording mistakes.
  • One limitation of ecological studies is the potential for bias inherent to the study design. This is a limitation of the current study as ecological studies use population level data instead of individual level data. I recommend discussing this limitation in the discussion of the manuscript.
  • In the discussion section, cafestol and kahweol are mentioned as potential mediators for the observed protective effect. However, both are known to be low in concentration in filtered coffee, which is the most used type in most of the countries included in the study (https://doi.org/10.3390/ijms20174238).
  • Causality and reverse causality are out of scope of ecological and observational studies except for Mendelian randomization analysis. Therefore, all causality claims can not be supported by the study’s results.

Reviewer 2 Report

Brief summary

The short communication “Coffee Consumption and its Inverse Association with Gastric Cancer: An Observational Study” quantifies the relationship between coffee consumption and the incidence and mortality of gastric cancer in 22 countries using an ecological study approach. An inverse linear correlation between coffee consumption kg per person per year and estimated age-adjusted incidence in morality of gastric cancer was observed.

Comments and suggestions for authors:

English changes are required: There are many grammatical and typographical errors throughout the document. I suggest the article be reviewed by the authors again as well as reviewed by a native English speaker.

Minor editing throughout the manuscript: avoid the use of “we” in the Introduction, Materials and Methods, and Results; review the use of abbreviations; be consistent with writing out numbers <10…

Title

Although the current short communication is of observational nature, it may be more informative to add that it is an ecological study to the title.

Introduction

The introduction does fully explore the results of previous studies that have examined the association between coffee consumption and gastric cancer risk.

Methods

Could information on coffee availability be retrieved from the FAO of the UN Food Balance Sheets in order to increase the number of countries included in the study?

Results

Perhaps define the correlation as moderate or strong, as applicable, in the results.

The results do not mention the outliers such as Portugal, Estonia, Bosnia and Herzegovina, and USA.

Figure 1 and 2 consider adding each country’s flag.

Discussion

The authors state that available evidence on the association between coffee consumption and gastric cancer is inconclusive but do not explain or discuss why these differences are observed.

The authors do not discuss the outliers such as Portugal, Estonia, Bosnia and Herzegovina, and USA.

Reverse causation is only discussed regarding gastric cancer mortality and not gastric cancer incidence.

There is no discussion of potential confounders such as socioeconomic characteristics, other lifestyle factors (fruit and vegetable intake, red meat intake, salt consumption, smoking and alcohol intake), infection H. pylori which all vary from country to country.

Another limitation of the study, besides “methods of coffee preparation (espresso, boiled, or filtered) and coffee seeds (Arabica or Canephora) that was not available for our analyses” would be the lack of information on the temperature of the coffee.

There is no discussion of one of the main limitations of ecological studies: the unit of observation is the population level (country) and not the individual.

Round 2

Reviewer 1 Report

Dear Authors, 

I would like to thank the authors for their extensive work on their manuscript. I believe the quality and readability of the mansucript increased significantly.

I only have one minor note regarding the line 176. Mendelian randomization is an analysis approach to assess causality in observational studies.

Kind regards

Reviewer 2 Report

The authors have successfully incorporated all changes into the manuscript.

Thank you.